# Evaluation of Clinical Biomarkers Related to CD4 Recovery in HIV-Infected Patients—5-Year Observation

**DOI:** 10.3390/v14102287

**Published:** 2022-10-18

**Authors:** Agnieszka Lembas, Andrzej Załęski, Tomasz Mikuła, Tomasz Dyda, Wojciech Stańczak, Alicja Wiercińska-Drapało

**Affiliations:** 1Department of Infectious and Tropical Diseases and Hepatology, Medical University of Warsaw, 02-091 Warsaw, Poland; 2Hospital for Infectious Diseases in Warsaw, 01-201 Warsaw, Poland; 3Molecular Diagnostics Laboratory, Hospital for Infectious Diseases, Warsaw, 01-201 Warsaw, Poland

**Keywords:** HIV, CD4, CD4:CD8, antiretroviral therapy, immune reconstruction

## Abstract

Human Immunodeficiency Virus infection leads to the impairment of immune system function. Even long-term antiretroviral therapy uncommonly leads to the normalization of CD4 count and CD4:CD8 ratio. The aim of this study was to evaluate possible clinical biomarkers which may be related to CD4 and CD4:CD8 ratio recovery among HIV-infected patients with long-term antiretroviral therapy. The study included 68 HIV-infected patients undergoing sustained antiretroviral treatment for a minimum of 5 years. Clinical biomarkers such as age, gender, advancement of HIV infection, coinfections, comorbidities and applied ART regimens were analyzed in relation to the rates of CD4 and CD4:CD8 increase and normalization rates. The results showed that higher rates of CD4 normalization are associated with younger age (*p* = 0.034), higher CD4 count (*p* = 0.034) and starting the therapy during acute HIV infection (*p* = 0.012). Higher rates of CD4:CD8 ratio normalization are correlated with higher CD4 cell count (*p* = 0.022), high HIV viral load (*p* = 0.006) and acute HIV infection (*p* = 0.013). We did not observe statistically significant differences in CD4 recovery depending on gender, HCV/HBV coinfections, comorbidities and opportunistic infections. The obtained results advocate for current recommendations of introducing antiretroviral therapy as soon as possible, preferably during acute HIV infection, since it increases the chances of sufficient immune reconstruction.

## 1. Introduction

Human Immunodeficiency Virus (HIV) infection affects more than 38 million people in the world. In 2021, there were approximately 1.5 million new HIV-infection diagnoses [1]. In 2020, among all newly diagnosed patients, more than 50% were late presenters, with CD4 counts below 350 cells per mm^3^, and more than 30% from the whole cohort had CD4 counts below 200 cells per mm^3^ [2].

Untreated HIV infection leads to the progressive and continuous impairment of the immune system function [3]. The successive loss of peripheral blood CD4+ T cells can result in the development of opportunistic infections [4]. However, effective antiretroviral therapy (ART) can prevent the drop in CD4 count and to some extent restore the CD4 cell level [5]. Current studies suggest that people living with HIV (PLWH) who are undergoing antiretroviral treatment have similar life expectancy to the non-infected population, despite the higher incidence of comorbidities [6].

The CD4 cell count ≥ 500 cells/µL and CD4:CD8 ratio ≥ 1 are currently considered normal range and remain the target of immune reconstitution in HIV-infected patients [7]. Patients with a persistence of lower CD4 and CD4:CD8 ratios despite antiretroviral therapy were named as inadequate immunological responders or immunological non-responders (INRs). Being an immunological non-responder may result in increased risk of progression to AIDS and non-AIDS events, and therefore higher rates of mortality [8]. The mechanisms for this phenomenon may be multifactorial [9]. Clinical risk factors for impaired CD4 recovery have not been established; however, older age, male gender, low CD4 cell count and low CD4:CD8 ratio at the time of diagnosis have been associated with worse immunological response to antiretroviral treatment [10].

Since the reasons for diverse immune reconstruction in HIV-infected patients are still not fully known and explained, we aimed to investigate this matter. We believe it is vital to search for clinical biomarkers which could be the predictors of immunologic reconstitution. Knowing these could possibly increase the chances of enhanced immune reconstruction. Moreover, it could help clinicians identify those patients at risk of maintaining low CD4 cell counts and CD4:CD8 ratios and extend the supervision over them in terms of the progression of HIV infection and development of opportunistic infections. We could also identify individuals being at risk of non-AIDS events and provide multi-specialized care to prevent this. For those reasons, our study aimed to evaluate possible clinical biomarkers which may be related to CD4 cell count and CD4:CD8 ratio recovery among HIV-infected patients with sustained long-term antiretroviral therapy.

## 2. Materials and Methods

### 2.1. Patients

The population of HIV-infected adult patients admitted to our department was analyzed in the period 2011–2022. The inclusion criteria were: HIV infection, persistent ART therapy for 5 years from the time of onset, reaching undetectable HIV viral load after a maximum of 1 year since the initiation of antiretroviral treatment and aged 18 or older. The exclusion criteria were the history of discontinuing ART at any time and any detectable HIV viral load after 1 year since beginning antiretroviral therapy.

### 2.2. Assessments

The analyzed patients underwent physical examination and laboratory testing. All patients were assessed in terms of HCV/HBV coinfections, comorbidities and opportunistic infections. The schemes of antiretroviral treatment applied during a 5-year observation period were studied. Blood samples were repetitively collected for CD4 count, CD4:CD8 ratio and HIV viral load. The first post-baseline examination was performed between 3 and 6 months after the introduction of ART, the second detection was after one year since the beginning of ART and the next examinations were performed yearly.

The immunophenotyping analyses included measurement of the absolute count of T lymphocyte (CD3+) subsets: CD4+ (helper/inducer), CD8+ (suppressor/cytotoxic) and CD4:CD8 ratio were determined using flow cytometry method with the application of three-color direct immunofluorescence reagents—TriTEST™ (BD Biosciences, North Ryde, Australia). A volume of 50 μL of whole blood samples collected the same day for EDTA anticoagulant were incubated in the darkness with the addition of 20 μL of fluorescence-conjugated monoclonal antibodies labelled appropriately: CD4-fluorescein isothiocyanate fluorescein (FITC)/CD8-phycoerythrin (PE)/CD3-peridinin chlorophyll protein (PerCP). Each sample stained with reagents was mixed with microbeads in a TruCOUNT™ tube and prepared according to manufacturer lyse/no-wash procedure. Data were acquired and analyzed using BD MultiSET™ software on the multicolor, dual-laser BD FACSCalibur analyzer. After data collection from 15,000 events, a specific region was set on SSC-H low/CD3-PerCP high+ cells population considered to be T lymphocytes. The gating strategy was based on the selection of the appropriate area on two parameter histograms for distribution of cell populations due to labelled markers CD3-PerCP/CD4-FITC and CD4-FITC/CD8-PE, by using the software provided. The ratio of fluorescent cells to TruCOUNT beads multiplied by the known concentration of beads in the tube was automatically recalculated by the built-in algorithm to the CD3+, CD4+ and CD8+ T-lymphocytes as absolute numbers of lymphocytes per microliter of blood analyzed.

HIV viral load was assessed by the Abbott RealTime HIV-1 assay using an in vitro reverse transcription-polymerase chain reaction (RT-PCR) assay with homogenous real-time fluorescent detection for the quantitation of Human Immunodeficiency Virus type 1 (HIV-1) on the automated m2000 System in human plasma. The assay used RT-PCR26 to generate amplified product from the RNA genome of HIV-1 in clinical specimens. The amount of HIV-1 target sequence was measured through the use of fluorescent labeled oligonucleotide probes on the Abbott m2000rt™ instrument. The range of the performed test was 40 to 10,000,000 copies/mL. We adopted HIV viral load < 40 copies/mL as undetectable.

We analyzed CD4 cell count and CD4:CD8 ratio growth and normalization over the course of 5 years of sustained antiretroviral therapy. CD4 cell count ≥ 500 cells/µL and CD4:CD8 ratio ≥ 1 was considered the normal range according to CDC Guidelines for Performing CD4+ T-Cell Determinations in Persons Infected with Human Immunodeficiency Virus [7]. The patients with CD4 ≥ 500 cells/µL and CD4:CD8 ≥ 1.0 at baseline were included in the examination of CD4 cell count and CD4:CD8 ratio growth and excluded from the analysis of CD4 cell count and CD4:CD8 ratio normalization.

We assessed the change of CD4 cell count during a 5-year observation. We adopted the parameter ΔCD4 cell count which signifies the difference in baseline CD4 count and CD4 count after 5 years of sustained antiretroviral therapy, while ΔCD4:CD8 ratio signifies the difference in baseline CD4:CD8 ratio and CD4:CD8 ratio after 5 years of ART. In the tables, we presented the mean and standard deviation of ΔCD4 cell count and ΔCD4:CD8 ratio if the variable had a normal distribution or used the median and interquartile range if the variable did not have a normal distribution.

The study group characteristics, CD4 cell count increase (ΔCD4 cell count) and CD4:CD8 ratio increase (ΔCD4:CD8 ratio) were reported in means and minimum–maximum values (range). The data contained in the boxplots were reported in median, interquartile range and minimum–maximum values (range).

### 2.3. Statistical Analysis

The Shapiro–Wilk test was performed for the verification of the normality of the distributions in the analyzed variables. A Student’s t-test or Mann–Whitney U test were used to evaluate the difference in mean value in continuous variables, while χ2 or Fisher exact tests were performed for categorical variables. The Kruskal–Wallis ANOVA test was used to evaluate the difference in mean values among more than two quantitative variables. The *p*-value was set at 0.05. The analysis of variance for repeated measures with multiple factors and a generalized linear model with repeated measures showing the relationship between ΔCD4 cell count, ΔCD4:CD8 ratio, CD4 cell count normalization and CD4:CD8 ratio normalization and confounding factors was performed. All statistical analyses were performed using Python 3.7 software and the Statistica 13.1 program (StatSoft Poland, Kraków, Poland).

### 2.4. Ethics Approval

Ethical approval and written informed consent were waived by the Bioethics Committee of Medical University of Warsaw because of the retrospective nature of the study. Instead, the Bioethics Committee of Medical University of Warsaw approved the use of oral consent, which was documented in patients’ medical records. All analyzed patients’ data were fully anonymized. The study followed the principles of the Declaration of Helsinki.

## 3. Results

### 3.1. Study Group

We analyzed a group of 68 patients (61 men, 7 women) with diagnosed HIV infection. All patients had antiretroviral therapy introduced and continued uninterruptedly for a minimum of 5 years. At the beginning of antiretroviral treatment, 10 out of 68 patients (14.71%) had CD4 cell count ≥ 500 cells/µL and 2 out of 68 patients had CD4:CD8 ratio ≥ 1 (2.94%). The baseline characteristics of the patients are shown in Table 1.

Among patients, 8 had acute HIV infection, 8 had HCV coinfection, 3 had HBV coinfection and 22 had chronic comorbidities. Among the comorbidities were: hypercholesterolemia (*n* = 8), hypertension (*n* = 5), depression (*n* = 3), liver cirrhosis (*n* = 3), bipolar affective disorder (*n* = 1), Klinefelter syndrome (*n* = 1), chronic kidney disease (*n* = 1), Crohn’s disease (*n* = 1), ulcerative colitis (*n* = 1), type 2 diabetes (*n* = 1), atopic dermatitis (*n* = 1), adrenal insufficiency (*n* = 1), granulomatosis with polyangiitis (*n* = 1), sarcoidosis (*n* = 1) and Hodgkin’s lymphoma (*n* = 1).

In the analyzed group of patients, there were 25 people with AIDS-defining diseases. Among them, 11 people had more than one disease at the time: pneumocystis jirovecii pneumonia (*n* = 11), tuberculosis (*n* = 5), atypical mycobacterial disease (*n* = 5), esophageal candidiasis (*n* = 5), HIV encephalopathy (*n* = 3), CNS toxoplasmosis (*n* = 3), disseminated cytomegalovirus (CMV) disease (*n* = 2), Kaposi sarcoma (*n* = 2), cryptosporidiosis (*n* = 1) and cervical cancer (*n* = 1).

### 3.2. Antiretroviral Therapy

All of the analyzed patients were receiving a three-drug antiretroviral therapy regimen at some stage of the therapy. There most prevalent schemes were protease inhibitor-based therapy (one protease inhibitor (PI) plus two nucleotide analog reverse transcriptase inhibitors (NRTI)), non-nucleotide analog reverse transcriptase inhibitor-based therapy (one non-nucleotide analog reverse transcriptase inhibitor plus two nucleotide analog reverse transcriptase inhibitors) and integrase inhibitor-based therapy (one integrase inhibitor (InSTI) plus two nucleotide analog reverse transcriptase inhibitors (NRTI) or one integrase inhibitor plus one nucleotide analog reverse transcriptase inhibitor).

Among the cohort of patients, 33 individuals underwent the regimen change during a 5-year observation period (29 people underwent one regimen change and 3 people had two regimen changes). By the change of regimen, we assumed switching between groups: InSTI-based therapy, PI-based therapy and NNRTI-based therapy. Switching the pharmaceuticals with the same class of antiretrovirals was not considered a change of regimen. Table 2 presents the number of patients treated with different antiretroviral regimens.

### 3.3. CD4 Recovery in All Patients

CD4 cell count recovery (ΔCD4) among all patients during the 5-year observation period after the introduction of antiretroviral therapy is presented in Figure 1. The growth CD4:CD8 ratio growth (ΔCD4:CD8) among our cohort of patients is shown in Figure 2. In the analyzed group of patients, 41 out of 68 (60.29%) had CD4 cell count ≥ 500 cells/µL and 30 out of 68 (44.12%) had CD4:CD8 ratio ≥ 1 after 5 years of sustained antiretroviral treatment. Since 10 patients had CD4 cell count ≥ 500 cells/µL and 2 patients had CD4:CD8 ratio ≥ 1 at baseline, 21 out of 58 patients (36.21%) managed to reach CD4 cell count ≥ 500 cells/µL and 28 out of 66 patients (42.42%) managed to reach CD4:CD8 ratio ≥ 1 during 5 years of ART.

### 3.4. CD4 Recovery Depending on Clinical Biomarkers

We analyzed whether clinical biomarkers such as age, gender, baseline CD4 count, baseline HIV viral load, acute HIV infection, HCV/HBV coinfections, comorbidities and opportunistic infections impacted the rate of CD4 cell count recovery and CD4:CD8 ratio recovery. The results are presented in Table 3.

We investigated how many people belonging to each group have managed to reach CD4 cell count normalization. The results are shown in Table 4.

Age below 35 years old, high CD4 count at the beginning of antiretroviral therapy and acute HIV infection showed to be positive factors of CD4 cell count normalization in 5 years. In terms of CD4:CD8 ratio normalization during 5-year antiretroviral therapy, high baseline CD4 count, high HIV viral load and acute HIV infection appeared to have statistical significance. Figure 3 presents CD4 count and CD4:CD8 ratio recovery over 5 years of sustained ART.

We performed an analysis of variance for ΔCD4 cell count, ΔCD4:CD8 ratio, CD4 cell count normalization and CD4:CD8 ratio normalization. We obtained statistically significant *p* values for ΔCD4:CD8 ratio (*p* < 0.001), CD4 cell count normalization (*p* = 0.024) and CD4:CD8 ratio normalization (*p* = 0.016) and a statistically insignificant *p* value for ΔCD4 cell count (*p* = 0.072). A generalized linear model with repeated measures showing the relationship between ΔCD4 cell count, ΔCD4:CD8 ratio, CD4 cell count normalization and CD4:CD8 ratio normalization and confounding factors was also performed. For qualitative variables, we analyzed men vs. women, individuals with no acute HIV infection vs. patients with acute HIV infection, people with HCV/HBV coinfection vs. people without coinfections, patients with no comorbidities vs. patients with comorbidities and individuals with no opportunistic infections vs. people with the diagnosis of opportunistic infection. The results are presented in Table 5 and Figure 4.

In a generalized linear model with repeated measures, we also analyzed quantitative variables: baseline CD4 cell count, age of initiation of ART and HIV viral load at the start of antiretroviral therapy. We observed statistically significant *p* values for CD4 cell count increase in younger patients (*p* = 0.020) and lower baseline CD4 cell count (*p* < 0.001) and a statistically insignificant *p* value for HIV viral load (*p* = 0.696). In terms of CD4:CD8 ratio growth, we did not obtain statistically significant *p* values for age (*p* = 0.756), baseline CD4 count (*p* = 0.628) and HIV viral load (*p* = 0.939).

### 3.5. CD4 Recovery Depending on the Regimens of Antiretroviral Therapy

We analyzed three groups of patients depending on the applied antiretroviral therapy regimen. The first group of patients were undergoing protease inhibitor-based therapy (2 NRTI + 1 PI), the second group were receiving integrase inhibitor-based therapy (2 NRTI + 1 InSTI or 1 NRTI + 1 InSTI) and the third group were having non-nucleotide analog reverse transcriptase inhibitor-based therapy (2 NRTI + 1 NNRTI or 2 NRTI + 2 NNRTI). Table 6 presents CD4 cell count and CD4:CD8 ratio recovery among patients receiving different antiretroviral regimens. Table 7 shows the number of people who reached CD4 cell count and CD4:CD8 ratio normalization among patients receiving different antiretroviral regimens.

We did not observe statistically significant differences in the CD4 cell count and CD4:CD8 ratio increase and in the number of people who have managed to reach CD4 cell count and CD4:CD8 ratio normalization depending on applied antiretroviral treatment.

We presented the CD4 count and CD4:CD8 ratio recovery over 5 years in three groups of patients: receiving InSTI-based therapy, PI-based therapy and NNRTI-based therapy in Figure 5.

We also evaluated whether the changes in the antiretroviral treatment regimen had an impact on CD4 cell count and CD4:CD ratio increase and CD4 count and CD4:CD8 ratio normalization. The results of CD4 cell count and CD4:CD ratio increase are shown in Table 8. The results of CD4 cell count and CD4:CD8 ratio normalization are presented in Table 9.

There were no significant differences in CD4 cell count increase, CD4:CD8 ratio increase, CD4 count normalization and CD4:CD8 ratio normalization among patients who had the antiretroviral therapy regimen changed and among those who had one scheme applied.

## 4. Discussion

Immune reconstruction in HIV-infected patients is a process of rebuilding the immune system after the introduction of combined antiretroviral therapy. Centers for Disease Control and Prevention (CDC) consider CD4 cell count ≥ 500 cells/µL and CD4:CD8 ratio ≥ 1 as the normal range and one of the goals of HIV treatment [7]. Immune system recovery is most important in an advanced stage of the disease due to the increased risk for the development of opportunistic infections and neoplasms [4]. Studies show that even long-term antiretroviral treatment uncommonly leads to the normalization of CD4 count and CD4:CD8 ratio [11,12]. There are reports suggesting that CD4:CD8 ratio may better reflect immune dysfunction in well-controlled HIV infection than CD4+ cell count alone [13,14].

Among patients included in our study, 58 (85.29%) had CD4 cell count < 500 cells/µL and CD4:CD8 ratio < 1 at the initiation of ART. In those people, 36.21% managed to reach CD4 cell count ≥ 500 cells/µL and 42.42% gained CD4:CD8 ratio ≥ 1 during 5 years of ART. Among 46 patients (67.65%) with CD4 cell count < 350 cells/µL, 22 (47.83%) managed to reach CD4 ≥ 500 cells/µL and 16 (34.78%) gained CD4:CD8 ratio ≥ 1. Among 30 patients (44.12%) with CD4 < 200 cells/µL, 11 (36.67%) reached CD4 cell count normalization and 6 (20.00%) gained CD4:CD8 ratio normalization. In a study with a 4-year follow-up, 44% of patients normalized CD4 cell count and only 33% normalized CD4:CD8 ratio [15]. In other reports, the follow-up was shorter than 5 years. There was a study conducted in which during the median 2.6 years of observation, 28% of patients normalized CD4:CD8 ratio [16]. In another study with a median observation period of 2.77 years, only 7.2% of people normalized CD4:CD8 ratio [17]. Mussini et al. estimated the probability of ratio normalization during 5 years of ART and obtained a result of 29.4% [18]. That may suggest that longer antiretroviral treatment is associated with higher CD4:CD8 ratio normalization rates.

In our study, we analyzed whether various factors may have an impact on CD4 and CD4:CD8 ratio recovery. We evaluated biomarkers such as age, gender, CD4 cell count and CD4:CD8 ratio at the beginning of the therapy, HIV viral load at the beginning of ART, the existence of acute HIV infection, HCV or HBV coinfections, comorbidities or opportunistic infections.

### 4.1. CD4 Recovery Depending on Patient’s Age

We analyzed two groups of HIV-infected patients: those younger than 35 years old and those who were 35 years old or older at the start of antiretroviral treatment. We observed that there were significantly more people who managed to reach CD4 cell count ≥ 500 cells/µL in the group of younger patients. Apart from that, CD4:CD8 ratio normalization was also attained by more people in the younger patients’ group; however, that difference did not appear statistically significant.

There are studies which seem to support the thesis that younger age may be a positive predictor in terms of CD4 recovery. Yu et al., who also compared CD4 growth among patients ≥35 years old and <35 years old, obtained similar results: younger patients reached higher CD4 cell count during long-term antiretroviral treatment than older patients [19]. Chen et al. also evaluated that older patients (≥50 years old) gained lower median maximal CD4 cell count on antiretroviral therapy than patients who were younger than 50 years old [20]. There are more studies supporting that hypothesis [21]. There is also research involving children and adolescents with HIV infection indicating that in in this age group, older age is also associated with slower CD4:CD8 ratio recovery [22,23].

In contrast, there are also reports saying that age is not associated with CD4 recovery. An African cohort that included almost 3000 people living with HIV did not show differences in CD4 recovery among patients ≥50 and <50 years old when they started ART [24].

### 4.2. The Impact of Baseline CD4 Count on CD4 Recovery

Among the analyzed individuals, the proportion of patients with CD4 cell count < 350 cells/µL at the beginning of antiretroviral therapy accounted for 67.65% of the population, which is a high rate of late diagnoses. However, our study was conducted among patients diagnosed since 2011 and treated for a minimum of 5 years. In Poland before 2016, the antiretroviral treatment was not widely available to all HIV-infected patients [25]. Moreover, the study group might be biased, since the study was conducted among patients diagnosed in the hospital’s department, usually constituting more advanced stages of the infection than patients diagnosed in the outpatients’ clinic.

In our study, we analyzed CD4 recovery among patients with CD4 cell count ≥ 200 cells/µL and <200 cells/µL and also with CD4 ≥ 350 cells/µL and <350 cells/µL at the beginning of ART. Significantly, more patients with CD4 cell count ≥ 200 cells/µL obtained CD4 cell count and CD4:CD8 ratio normalization than patients with CD4 < 200 cells/µL. In the comparison of patients with CD4 ≥ 350 cells/µL and <350 cells/µL at ART initiation, we also observed a similar relationship; however, statistical significance was obtained only in terms of CD4:CD8 ratio normalization.

Other studies also seem to indicate that higher baseline CD4 cell count is associated with higher rates of patients who gain CD4:CD8 ratio normalization. There are researchers suggesting that starting ART with CD4:CD8 ratio > 0.5 is related with a greater likelihood of normalizing CD4:CD8 ratio [26]. Another study shows that significantly more people with baseline CD4 count < 100 cells/µL did not reach CD4 count ≥ 500 cells/µL in more than 6 years of observation [27]. A similar relationship of higher baseline CD4 count and CD4 recovery acceleration is also found in children [23].

On the contrary, there are studies which indicate that lower baseline CD4 cell count is associated with higher potential for improvement; therefore, patients with lower baseline CD4 count have a greater rate of CD4 count recovery [28]. That study contained 10 years of observation, so it is possible that patients with low baseline CD4 count need more time for CD4 recovery. The observation of our cohort continues.

### 4.3. CD4 Recovery in Patients Who Start ART with Acute HIV Infection

In our cohort, there were 8 patients who had antiretroviral therapy introduced during acute HIV infection. Among them, 7 individuals (87.50%) had CD4 cell count < 500 cells/µL and CD4:CD8 ratio < 1 at the point of HIV infection diagnosis. Those 7 patients all achieved CD4 cell count normalization within 5 years of ART. During the observation, 6 out of 7 patients (85.71%) also managed to reach CD4:CD8 ratio normalization. The rates of CD4 and CD4:CD8 normalization in patients diagnosed in later stages of the disease were much lower (47.06% and 33.33% of patients, respectively). Those differences appear statistically significant.

There are many studies which state that early initiation of ART in HIV-infected patients is very important. Researchers confirm that antiretroviral treatment introduced in acute HIV infection is effective in CD4 cell count recovery [29] and has the beneficial role in CD4 cells recovery and rates of CD4:CD8 ratio normalization [30]. Apart from CD4 recovery, early ART enables the clearance of other cells infected by HIV, such as in gut-associated lymphoid tissue and lymph nodes [31]. Early ART helps in bone marrow and peripheral B cells recovery, which are also affected by HIV infection [32,33].

There are also reports suggesting that immune recovery is comparable in primary and chronic HIV infection [34] or that suboptimal CD4 recovery occurs uncommonly evenly when introducing ART in acute HIV infection [35]. However, those studies also highlight the importance of early initiation of the treatment, since it is beneficial.

### 4.4. HIV Viral Load and CD4 Recovery

Our analyses show that HIV viral load ≥ 1 million copies/mL at the beginning of antiretroviral therapy, which concerned 16 patients (23.53%), was associated with higher rates of CD4:CD8 ratio normalization. We did not observe a similar relationship concerning CD4 cell count normalization. Among analyzed patients, similar to other studies, there were both patients with acute HIV infection (5, 31.25%) and advanced HIV infection (11, 68.75%). There are little data concerning the impact of HIV viral load on CD4 recovery; however, there are some studies which seem to support our results. Muscatello et al. demonstrated that there may be a relationship between higher HIV viral load and CD4 recovery [36]. In that study, high HIV viral load at ART initiation was associated with acute HIV infection, and that may be the reason for the obtained results. Another study, which did not analyze patients with acute HIV infection, connected high viral load with an increased risk of AIDS and, thus, inferior rates of CD4 recovery [37]. Since our population of patients with high HIV viral load included both patients with acute HIV infection and patients with advanced HIV infection, the results are difficult to compare with those of other researchers.

### 4.5. CD4 Recovery Depending on Patient’s Gender

Our study involved 68 HIV-infected patients, with 61 patients male and 7 female. We did not obtain statistically significant results, likely due to the inequality of the two groups. We observed that during 5 years of antiretroviral treatment, CD4 cell count and CD4:CD8 ratio increased more among women than men (the mean ΔCD4 among women was 476.43 cells/µL vs. men 320.00 cells/µL, while the mean ΔCD4:CD8 among women was 1.11 vs. men 0.66). Moreover, a larger percentage of women reached CD4:CD8 ratio normalization after 5 years (50.00% of women, 38.46% of men). We did not observe that relationship in CD4 cell count normalization (50.00% of women, 53.85% of men).

There are few current studies concerning the influence of gender on CD4 recovery. A French study reports that being a female may have beneficial impact on immune recovery, especially during long-term antiretroviral treatment, which may give women additional protection from adverse clinical events and premature ageing [38]. Another study suggested that there are no significant differences in CD4 recovery among men and women in chronic HIV infection [39]. Meditz et al. reported similar rates of CD4 cell count growth between men and women, noting that women had an elevated risk of HIV/AIDS-related events [40]. Moreover, there are studies suggesting that women’s poor CD4 recovery may be associated with vitamin D insufficiency [41].

### 4.6. The Influence of HCV/HBV Coinfection on CD4 Normalization

In our study group, 8 individuals were coinfected with HCV and 3 with HBV. The increase in CD4 cell count and CD4:CD8 ratio during 5 years of ART was higher among patients without HCV or HBV infection than in individuals with coinfections. The rates of CD4:CD8 normalization were also higher in the group of patients without HCV/HBV coinfection; however, rates of CD4 cells normalization were greater in the population of patients with coinfections. These results were not statistically significant.

Different studies suggest that HCV coinfection may have a negative impact on CD4 recovery among HIV/HCV-coinfected patients [42]. HCV may accelerate the depletion of CD4 cells because of the accumulation of dysfunctional immune activation during chronic viral infection [43]. Moreover, studies show that the introduction of direct-acting antiviral agents (DAA) leads to rapid decrease of CD4 cell count at the beginning of DAA therapy. The decrease may last even after the achievement of sustained virological response (SVR), but HCV clearance may induce improvement [44]. On the other hand, there are studies suggesting that HCV clearance after DAA treatment does not seem to have an impact on CD4 cell recovery [45]. On the contrary, researchers suggest that in HIV-infected children who are also vertically infected with HCV, HCV coinfection does not have a negative effect in long-term CD4 recovery compared to children with HIV infection alone [46].

Researchers studying HBV/HIV coinfection suggest that the presence of HBcAb seems to be associated with reduced CD4:CD8 ratio growth [47]. Another study suggests that in HIV/HBV-coinfected HBeAg-negative patients, immune recovery is continuously lower than in HBeAg-positive and HIV-mono-infected individuals [48]. One African study showed the acceleration of CD4 cell count recovery in HBV/HIV-coinfected patients with high HBV DNA viral load after ART initiation; however, it did not lead to increased rates of CD4:CD8 ratio normalization [49]. In children with HIV/HBV coinfection, similarly to HCV infection, CD4 increase was similar than in patients with HIV infection alone [50].

In general, recent studies suggest that in patients with HIV infection coinfected with HCV or HBV, CD4 recovery rates are lower than in patients with HIV mono-infection [51].

### 4.7. Comorbidities and CD4 Recovery

Among 68 HIV-infected patients in our study group, 22 (32.35%) were also diagnosed with other comorbidities. The most common chronic illnesses were: hypercholesterolemia, hypertension, depression and liver cirrhosis. Patients with comorbidities had slightly lower rates of CD4 cell count and CD4:CD8 ratio normalization; however, those differences were not statistically significant.

There are little data describing the impact of comorbidities on CD4 recovery in HIV-infected patients. Some studies suggest that dyslipidemia in HIV-infected patients may result in worse CD4 recovery outcomes [52,53]. Another study shows that high levels of HDL particles, HDL cholesterol and larger sizes of LDL particles have a better CD4 recovery than patients with high ratios of non-HDL lipoprotein particles [54]. Moreover, the administration of both statins and fibrates in HIV-related dyslipidemia do not seem to act significantly on clinical immune response in patients receiving antiretroviral treatment [55]. Hypertension, together with CD4 recovery, is suggested to be an epiphenomenon of the improvement of the HIV infection state, not the influencing factor [56]. Depression seems to be a risk factor for incomplete short-term HIV viral suppression among HIV-infected patients, and therefore poor CD4 cell count recovery [57]. Little is known whether liver cirrhosis affects CD4 cell count recovery; however, there are studies indicating that HIV/HCV-coinfected patients with lower CD4 recovery rates show more intense destructive processes in the liver than successfully recovered subjects [58], while also higher rates of CD4 recovery may lead to transient liver injury in patients with HIV/HCV coinfection, due to activation of the immune process [59].

### 4.8. The Impact of the Presence of AIDS-Defining Diseases on CD4 Recovery

In our study group, there were 25 patients with at least one opportunistic infection. We compared their rates of CD4 recovery to patients with Acquired Immunodeficiency Syndrome (AIDS) without opportunistic infections (CD4 cell count < 200/µL—7 patients). We observed lower CD4 cell count and CD4:CD8 ratio normalization rates among patients in the first group during 5 years of antiretroviral treatment (CD4 cell count normalization: 34.78% vs. 42.86%, respectively; CD4:CD8 ratio normalization: 13.04% vs. 42.86%, respectively). These differences were not statistically significant, probably due to the small sample size.

The available data examining the impact of the occurrence of opportunistic infections on CD4 recovery after antiretroviral therapy introduction are lacking. There are studies indicating that the development of AIDS is associated with poorer rates of CD4 and CD4:CD8 normalization than among patients diagnosed in earlier stages of HIV infection [10,60]. Many researchers acknowledge the importance of immunological non-responders (INR)—who are HIV-infected individuals failing to achieve the normalization of CD4 cell counts despite persistent virological suppression. That phenomenon may concern even up to 40% of people living with HIV [61]. These patients have an increased risk of progression to AIDS and non-AIDS events and present higher rates of mortality than HIV-infected individuals with adequate immune reconstitution [62]. The risk factors for INR are lower nadir CD4 T cell count, lower CD4:CD8 ratios and a lower naïve/memory CD4 cell ratio [58]. The predictor of long-term immunologic recovery in advanced HIV patients can be the CD4 slope during the first year of antiretroviral treatment [63]. Our study showed a plateau of CD4 cell count and CD4:CD8 ratio after approximately 1 year of antiretroviral treatment.

### 4.9. Applied Regimen of ART Influencing CD4 Recovery

Our analyses included patients hospitalized in 2011–2022; therefore, it is difficult to assess the therapeutic approach in all individuals. During the 11-year period, the therapeutic options were expanding and the recommendations regarding introducing various medications were changing. Thus, we decided to analyze the treatment regimens by the groups of applied therapeutics. The prevalent ART regimens in analyzed patients were: protease inhibitor-based therapy (47 patients), integrase inhibitor-based therapy (23 patients) and non-nucleotide analog reverse transcriptase inhibitor-based therapy (31 patients). There were no statistically significant differences between ΔCD4 and ΔCD4:CD8 among patients belonging to these three groups. The CD4 and CD4:CD8 normalization rates between patients in these groups also did not reach statistical significance; however, in the group of patients treated with integrase inhibitor-based therapy, more individuals managed to gain CD4:CD8 normalization rates than in other groups. We also examined whether the changes of regimen had an impact on CD4 recovery, but did not uncover statistically significant differences between patients who underwent one or two changes of regimen and individuals who did not change the scheme of the therapy.

Some studies also suggest that InSTI-based regimens show a better immune recovery rate and the type of first-line ART can have an impact on immune reconstitution [64,65]. One study compared the efficacy of regimens including raltegravir and efavirenz in CD4 recovery, with raltegravir seeming to lead to faster CD4:CD8 ratio normalization [66]. It is probable that this relationship may exist only when introducing antiretroviral treatment in advanced HIV infection. When ART was introduced in acute or recent HIV infection, viral suppression and immunological recovery were excellent, with no differences between ART regimens [67]. The meta-analysis including 33 studies showed that there were no effective medications specific for improving CD4 cell count reconstitution [68].

### 4.10. Limitations of the Study

The main limitation of our study was a small study population. Some analyses included a differentiated number of individuals because of the retrospective nature of the study and the available data. Therefore, in a few analyses, it is difficult to obtain statistically significant results. Moreover, we did not perform the stratification of therapeutic regimens by the applied drugs. We also did not perform the calculation and justification of the sample size selected.

## 5. Conclusions

Our study, which evaluated CD4 recovery during 5 years of effective antiretroviral treatment, suggests that clinical biomarkers such as younger age, higher CD4 baseline cell count, higher HIV viral load at the initiation of ART and the introduction of ART at the point of acute HIV infection are positive predictors of immune reconstitution. We therefore advocate for current recommendations to introduce antiretroviral therapy as soon as possible, preferably during acute HIV infection, since it provides the highest rates of CD4 cell count and CD4:CD8 ratio normalization. To make this possible, it is valid to introduce common rapid antiretroviral therapy in all HIV-infected patients.

## Figures and Tables

**Figure 1 viruses-14-02287-f001:**
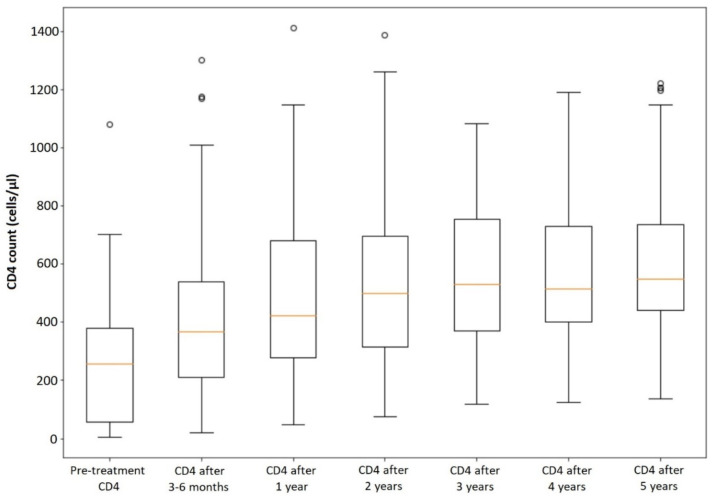
CD4 count recovery among all patients. The hollow circle stands for the outlier.

**Figure 2 viruses-14-02287-f002:**
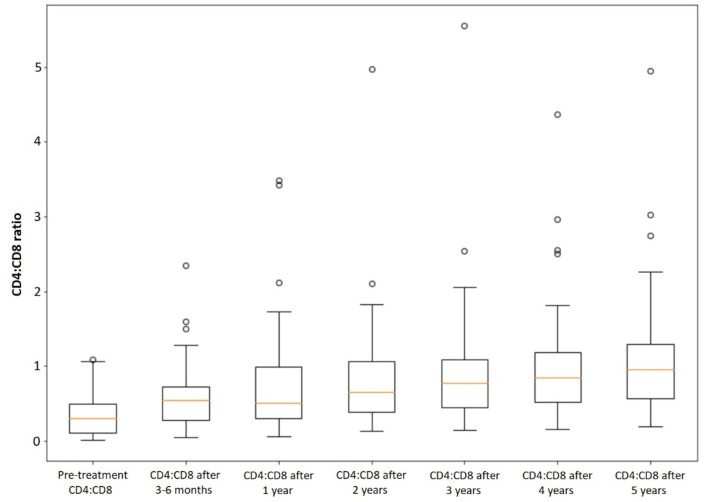
CD4:CD8 ratio growth among all patients. The hollow circle stands for the outlier.

**Figure 3 viruses-14-02287-f003:**
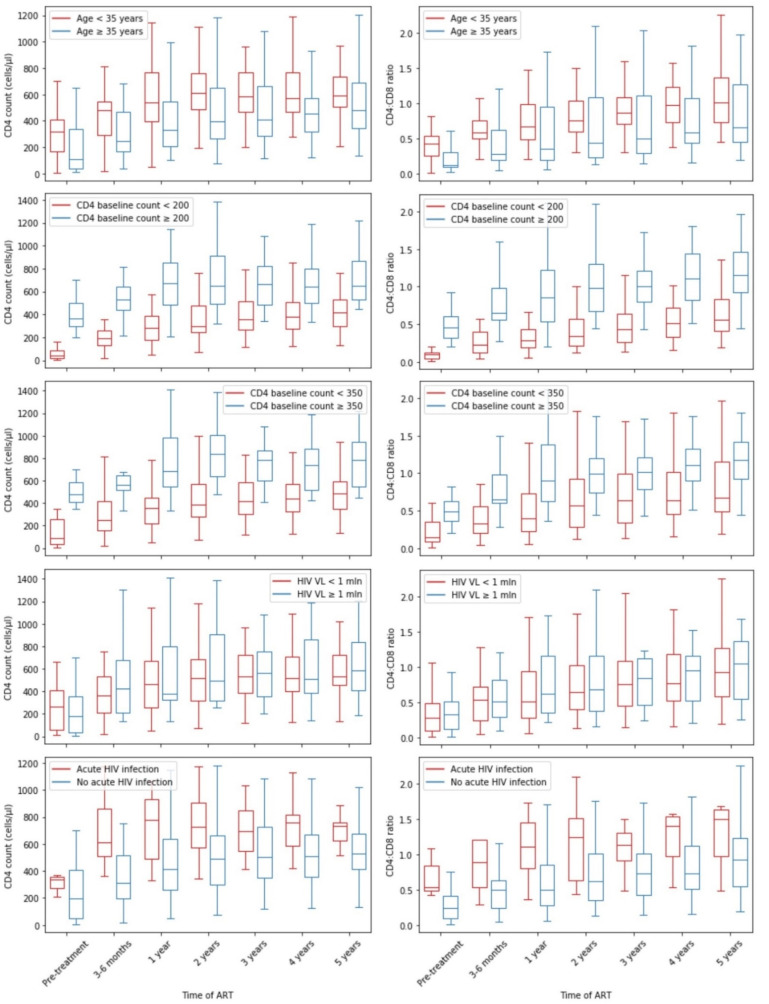
CD4 and CD4:CD8 recovery during 5 years of ART depending on clinical biomarkers.

**Figure 4 viruses-14-02287-f004:**
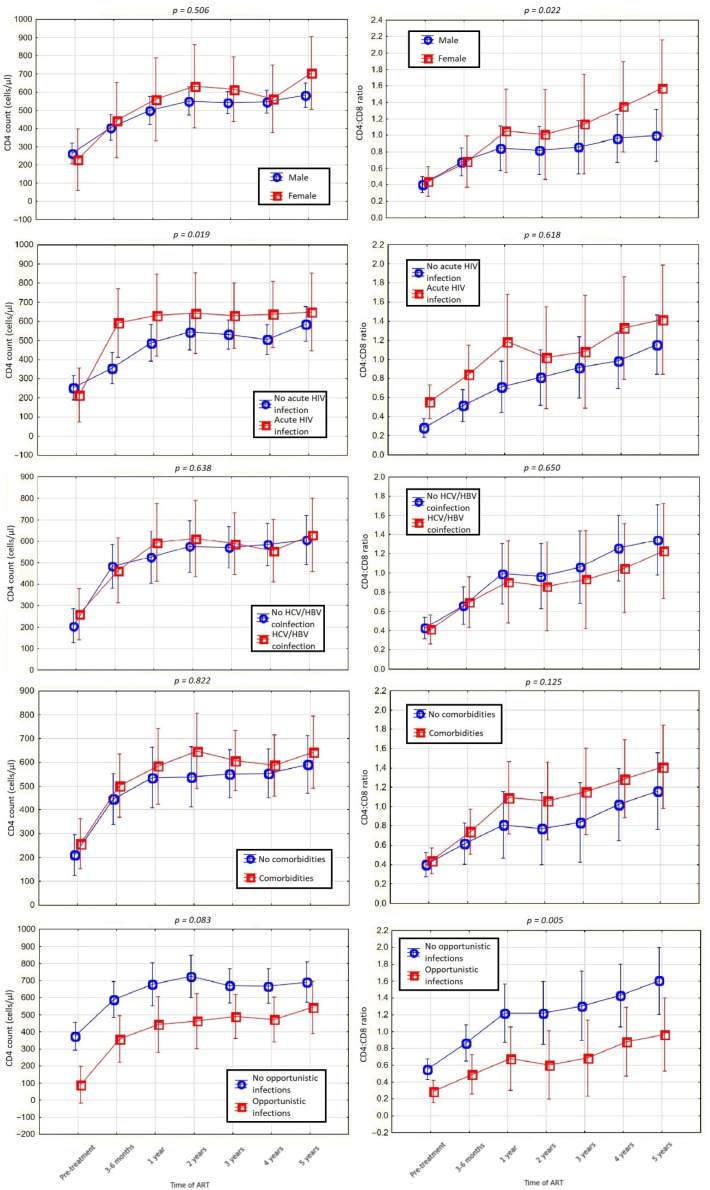
The results of a generalized linear model with repeated measures for CD4 cell count and CD4:CD8 ratio.

**Figure 5 viruses-14-02287-f005:**
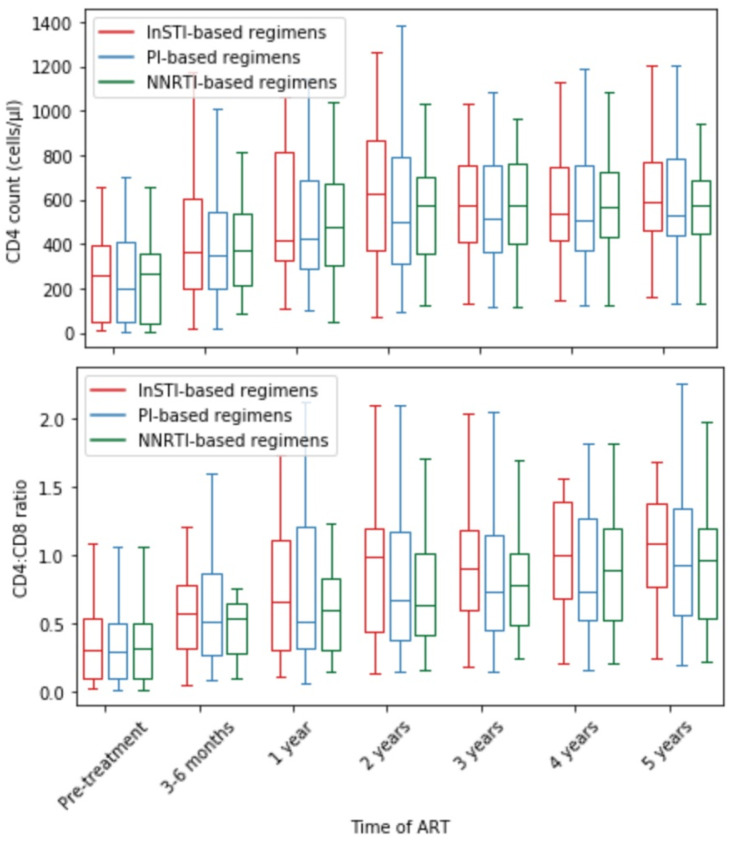
CD4 cell count and CD4:CD8 recovery depending on the applied ART regimen.

**Table 1 viruses-14-02287-t001:** Study group characteristics.

Variable (*n* = 68)	Mean (Range)
Age	
Age of HIV diagnosis (years)	36.21 (18–65)
Age of initiation of ART (years)	36.78 (18–65)
CD4 count and CD4:CD8 ratio	
CD4 count at the initiation of ART (cells/µL)	260.67 (5–1081)
CD4:CD8 ratio at the initiation of ART (proportion)	0.33 (0.01–1.09)
CD4 count after 5 years of ART (cells/µL)	596.93 (136–1223)
CD4:CD8 ratio after 5 years of ART (proportion)	1.04 (0.19–4.95)
HIV viral load	
HIV viral load at the initiation of ART (copies/mL)	1,526,393 (106–26,600,930)
HIV viral load after 5 years of ART (copies/mL)	undetectable

**Table 2 viruses-14-02287-t002:** Applied regimens of antiretroviral therapy.

Treatment Regimen	Occurrence *n* (%)
2 NRTI + PI	47 (69.12)
2 NRTI + NNRTI	31 (45.59)
2 NRTI + InSTI	18 (26.47)
1 NRTI + InSTI	7 (10.29)

**Table 3 viruses-14-02287-t003:** CD4 cell count and CD4:CD8 ratio recovery depending on clinical biomarkers.

Group of Patients	CD4 Cell Count Increase (ΔCD4 Cell Count)	*p*	CD4:CD8 Ratio Increase (ΔCD4:CD8 Ratio)	*p*
Age of initiation of ART < 35 (*n* = 33)	349.85 (247.74)	0.640	0.64 (0.41)	0.264
Age of initiation of ART ≥ 35 (*n* = 35)	323.14 (220.88)		0.50 (0.50)	
Male gender (*n* = 61)	320.00 (225.06)	0.093	0.54 (0.52)	0.097
Female gender (*n* = 7)	476.43 (271.45)		1.11 (0.94)	
CD4 < 200 cells/µL (*n* = 30)	347.50 (218.50)	0.087	0.43 (0.50)	0.018
CD4 ≥ 200 cells/µL (*n* = 38)	302.45 (246.30)		0.65 (0.50)	
CD4 < 350 cells/µL (*n* = 46)	339.00 (197.00)	0.082	0.53 (0.47)	0.181
CD4 ≥ 350 cells/µL (*n* = 22)	274.36 (282.96)		0.63 (0.74)	
HIV viral load < 1 mln copies/mL (*n* = 52)	319.48 (236.42)	0.292	0.54 (0.55)	0.284
HIV viral load ≥ 1 mln copies/mL (*n* = 16)	390.13 (219.58)		0.70 (0.56)	
Acute HIV infection (*n* = 8)	384.50 (53.00)	0.150	0.75 (0.74)	0.294
No acute HIV infection (*n* = 60)	331.13 (245.31)		0.54 (0.54)	
HCV/HBV coinfection (*n* = 12)	321.67 (268.91)	0.815	0.58 (0.36)	0.292
No HCV/HBV coinfection (*n* = 56)	339.20 (227.09)		0.58 (0.53)	
Any comorbidity (*n* = 22)	344.41 (239.01)	0.199	0.51 (0.59)	0.412
No comorbidities (*n* = 46)	311.5 (232.25)		0.62 (0.38)	
Any opportunistic infection and CD4 < 200 cells/µL (*n* = 23)	364.00 (216.00)	0.490	0.43 (0.49)	0.303
No opportunistic infection and CD4 < 200 cells/µL (*n* = 7)	354.00 (169.55)		0.59 (0.36)	

The gray numbers indicate mean (standard deviation); The bule number indicate median (interquartile range).

**Table 4 viruses-14-02287-t004:** CD4 cell count and CD4:CD8 normalization depending on clinical biomarkers.

Group of Patients	CD4 Count Normalization (≥500 Cells/µL) (*n*/%)	*p*	CD4:CD8 Ratio Normalization (≥1) (*n*/%)	*p*
Age of initiation of ART < 35 (*n* = 28)	20 (71.43%)	0.034	16 (57.14%)	0.063
Age of initiation of ART ≥ 35 (*n* = 30)	11 (36.67%)		7 (23.33%)	
Male gender (*n* = 52)	28 (53.85%)	1.000	20 (38.46%)	0.673
Female gender (*n* = 6)	3 (50.00%)		3 (50.00%)	
CD4 < 200 cells/µL (*n* = 30)	11 (36.67%)	0.034	6 (20.00%)	0.022
CD4 ≥ 200 cells/µL (*n* = 28)	20 (71.43%)		17 (60.71%)	
CD4 < 350 cells/µL (*n* = 46)	22 (47.83%)	0.115	16 (34.78%)	0.006
CD4 ≥ 350 cells/µL (*n* = 12)	9 (75.00%)		7 (58.33%)	
HIV viral load < 1 mln copies/mL (*n* = 45)	24 (53.33%)	1.000	16 (35.55%)	0.006
HIV viral load ≥ 1 mln copies/mL (*n* = 13)	7 (53.85%)		7 (53.85%)	
Acute HIV infection (*n* = 7)	7 (100.00%)	0.012	6 (85.71%)	0.013
No acute HIV infection (*n* = 51)	24 (47.06%)		17 (33.33%)	
HCV/HBV coinfection (*n* = 9)	6 (66.66%)	0.481	3 (33.33%)	0.686
No HCV/HBV coinfection (*n* = 49)	25 (51.02%)		20 (40.82%)	
Any comorbidity (*n* = 18)	9 (50.00%)	0.083	7 (38.88%)	0.061
No comorbidities (*n* = 40)	22 (55.00%)		16 (40.00%)	
Any opportunistic infection and CD4 <200 cells/µL (*n* = 23)	8 (34.78%)	1.000	3 (13.04%)	0.336
No opportunistic infection and CD4 < 200 cells/µL (*n* = 7)	3 (42.86%)		3 (42.86%)	

**Table 5 viruses-14-02287-t005:** The results of a generalized linear model for confounding factors for ΔCD4 cell count and ΔCD4:CD8 ratio and normalization.

Variable	CD4 Cell Count Increase (ΔCD4 Cell Count)		CD4:CD8 Ratio Increase (ΔCD4:CD8 Ratio)		CD4 Cell Count Normalization		CD4:CD8 Ratio Normalization	
	b’	*p*	b’	*p*	b’	*p*	b’	*p*
Age of initiation of ART	−0.247	0.082	−0.160	0.104	−0.331	0.018	−0.083	0.540
Gender	0.177	0.197	0.023	0.801	0.048	0.703	0.176	0.163
Baseline CD4 cell count	−0.286	0.089	0.342	0.003	0.212	0.220	0.077	0.651
HIV viral load	−0.077	0.677	0.151	0.196	−0.023	0.886	0.012	0.938
Acute HIV infection	0.185	0.310	0.259	0.030	0.290	0.084	0.238	0.150
HCV/HBV coinfection	−0.043	0.763	−0.017	0.854	0.099	0.439	−0.038	0.765
Comorbidities	0.066	0.665	0.044	0.644	0.029	0.824	−0.004	0.973
Opportunistic infections	0.136	0.449	−0.219	0.082	0.072	0.685	−0.331	0.065

b’—coefficient estimate on the scale of the linear predictor.

**Table 6 viruses-14-02287-t006:** CD4 cell count and CD4:CD8 recovery depending on the applied ART regimen.

Variable	PI-Based Therapy (*n* = 47)	InSTI-Based Therapy (*n* = 23)	NNRTI-Based Therapy (*n* = 31)	*p*
CD4 cell count increase	345.89 (228.52)	337.43 (265.16)	328.61 (254.71)	0.784
CD4:CD8 ratio increase	0.59 (0.61)	0.69 (0.51)	0.50 (0.51)	0.433

The gray numbers indicate mean (standard deviation); The bule number indicate median (interquartile range).

**Table 7 viruses-14-02287-t007:** CD4 cell count and CD4:CD8 normalization depending on the applied ART regimen.

Variable	PI-Based Therapy (*n* = 39)	InSTI-Based Therapy (*n* = 19)	NNRTI-Based Therapy (*n* = 26)	*p*
CD4 count normalization (≥500 cells/µL) (*n*/%)	21 (53.85)	12 (52.17)	13 (50.00)	0.251
CD4:CD8 ratio normalization (≥1) (*n*/%)	15 (38.46)	12 (52.17)	11 (42.31)	0.150

**Table 8 viruses-14-02287-t008:** CD4 cell count and CD4:CD8 recovery depending on changes in the ART regimen.

Variable	No Changes of Regimen (*n* = 36)	1 or 2 Changes of Regimen (*n* = 32)	*p*
CD4 cell count increase	319.94 (207.28)	354.28 (260.96)	0.159
CD4:CD8 ratio increase	0.59 (0.37)	0.62 (0.55)	0.209

The gray numbers indicate mean (standard deviation); The bule number indicate median (interquartile range).

**Table 9 viruses-14-02287-t009:** CD4 cell count and CD4:CD8 normalization depending on changes in the ART regimen.

Variable	No Changes of Regimen (*n* = 32)	1 or 2 Changes of Regimen (*n* = 26)	*p*
CD4 count normalization (≥500 cells/µL) (*n*/%)	17 (53.13)	14 (53.85)	0.564
CD4:CD8 ratio normalization (≥1) (*n*/%)	10 (31.25)	13 (50.00)	0.128

## Data Availability

Not applicable.

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
