# Peer review of "Evaluation of Clinical Biomarkers Related to CD4 Recovery in HIV-Infected Patients—5-Year Observation"

_viruses, 2022, doi:10.3390/v14102287_

Round 1

Reviewer 1 Report

The authors aim to evaluate possible clinical biomarkers which may be related to CD4 cell count and CD4:CD8 ratio recovery among HIV-infected patients with sustained long-term antiretroviral therapy.

The study design is appropriate but the analyzes performed are not adequate to the design. Many doubts and gaps are present in the methods, particularly in the statistical methods, and also in the results.

Major revisions

1) Has the verification of the normality of the distributions for the analyzed variables been carried out? What tests were used for the check on normality? There is no reference to this in the methods.

Is it not indicated how the statistics will be reported in the results: means, medians, standard deviations, interquatyl ranges?

2) The authors write that The ANOVA test was used to evaluate the difference in mean value among more than 2 quantitative variables. Which variables do they refer to? Parametric Anova or Kruskal-Wallis Non-Parametric Anova? It seems that a check on the normality of the variables has not been done.

3) Why Mann-Whitney U test and not Student's t-test was performed for continuous variables? If the variables had no normal distribution, was their transformation tested before opting for non-parametric tests?

4) Lines 137-138: It is not clear which values are reported in the brackets: range, confidence interval?

5) Figures 1 and 2: Figures 1 and 2 show the 3-6 month survey. What does it mean? The methods do not specify the detection times or explanations on the first post-baseline detection.

6) As these were repeated measures over time, an analysis of variance had to be made for repeated measures, with two or more factors.

Furthermore, to check the temporal trend of the two outcome measures between the levels of the covariates represented in Figures 3 and 4, a generalized linear model with repeated measures adjusted for possible confounding factors should be tested.

Minor revisions

1) Line 65: The authors write "repetitively", but do not indicate with what cadence.

2) Table 1: Enter the indication on the number of subjects.

3) Lines 138-143: The number of subjects who at baseline had values in the normal range has already been written previously, so you can directly report the numbers of those who have reached normalcy after 5 years.

4) Tables 3, 5, 7; it is not specified what the values shown in brackets refer to. The standard deviation or the interquartile range should be reported.

Author Response

Thank you very much for the substantive review. We took into consideration every suggestion and attempted to reply to all of them.

Major revisions
1) The manuscript was revised due to received suggestion and the verification of the normality of the distribution of the analyzed variables was performed. We used Shapiro-Wilk test. The study group characteristics, CD4 cell count increase (ΔCD4 cell count) and CD4:CD8 ratio increase (ΔCD4:CD8 ratio) was reported in means and mini-mum-maximum values (range). The data contained in the boxplots was reported in median, interquartile range and minimum-maximum values (range). The appropiate changes were done in the Methods section and in the tables.

2) We performed Shapiro-Wilk test for the verification of the normality of the distribution of the analyzed variables. In the analyses of 2 or more quantitative variables the groups were non-parametric, therefore we used Kruskal-Wallis Non-Parametric Anova. The appropiate changes were done in the Methods section and in the tables.

3) The manuscript was changed and Student's t-test was performed for continuous variables with normal distribution and Mann-Whitney U test was performed for groups without normal distribution. The appropiate changes were done in the Methods section and in the tables.

4) The values in lines 137-138 were mean and range (minimum-maximum). The appropiate changes were done in the Methods section and in the text.

5) In the methods section there were added the periods of follow-ups.

6) We performed an analysis of variance for repeated measures with multiple factors for ΔCD4 cell count, ΔCD4:CD8 ratio, CD4 cell count normalization and CD4:CD8 ratio normalization. A generalized linear model with repeated measures showing the relation between ΔCD4 cell count, ΔCD4:CD8 ratio, CD4 cell count normalization and CD4:CD8 ratio normalization and confounding factors was also performed. The appropiate changes were done in the text and Table 5 with statistics was added.

Minor revisions
1) We indicated the times of repetitive examinations in line 65.

2) We enter the indication on the number of subjects in Table 1.

3) Lines 138-143: We deleted the repeted text about the number of subjects who at baseline had values in the normal range.

4) We specified the values in brackets in Tables 3, 5, 7.

Reviewer 2 Report

Lembas and colleagues performed an observational study of clinical biomarkers for CD4 T cell recovery over five years in HIV infected individuals on antiretroviral treatment (ART). They find that age and CD4 count at time of ART initiation, as well as viral load and acute versus non-acute infection stage could significantly influence the recovery of CD4 T cell counts to 500/ul or above and the CD4:CD8 ratio to 1 or higher. The findings are in principle of great interest and the longitudinal design of the study is a major plus. As the authors acknowledge, the number of study participants (n=68) is small given the many variables related to ART, comorbidities, condition at start of ART and gender distribution. Thus the data and findings may be most interesting when put in context with other similar and future studies with more subjects. The authors could have used publish data from other studies to perform a preliminary and approximate power analysis. However, the main limitation for the full evaluation of this manuscript is the complete absence of information on the performed assays. In section 2.2 Assessments the authors need to describe what assays, incl. suppliers, for flow cytometry the antibodies and equipment used as well as the gating strategy. Also, how were viral titers determined (assay, supplier etc.)

Line 382 ADIS should read AIDS

Author Response

Thank you very much for the substantive review. We took into consideration every suggestion and attempted to reply to all of them.

We totally agree with you that the study group is small. That is due to the fact that we are a small HIV clinic and we included only patients who did not withdraw the treatment for minimum 5 years and regularly appeared for follow-ups. Unfortunately our study did not include data from other studies and including it now would mean a huge redesign of the whole study. However, we will continue delving into that topic and hopefully our next studies will contain power analysis also.

In section 2.2 Assessments we added more information about the performed assays of CD4 cell count and HIV viral load, including suppliers, antibodies, equipment used and diagnostic accuracy.

Line 382 the mistake in "AIDS" was corrected.

Reviewer 3 Report

Please correct some mistakes.

Line 153 please change CD4/CD4  for CD4/CD8

Line 193 CD4/CD8  add the 4 after CD

Line 207 add 8 after CD4/CD

Line 382 corrected AIDS

Author Response

Thank you very much for the positive review. We corrected the mistakes in lines: 153, 193, 207  and 382.

Reviewer 4 Report

Lembas et al. have reported on their experience with CD4+ cell reconstitution in a population of predominantly late presenters with HIV infection depending on various clinical and laboratory-based parameters. Although the manuscript is interesting, I still have a few suggestions on how it might be further improved before publishing is considered.

1.) Title: Regarding the high proportion of late presenters with AIDS-defining diseases within the study group, it might be misleading just to speak of “HIV-infections”. This high AIDS-proportion should already be declared in the title.

2.) In the abstract, it seems confusing why several markers are associated with more than one p-value. This section of the abstract should be rephrased to avoid this confusion.

3.) Material and methods, sub-heading “Assessments”: The description of the applied laboratory procedure seems quite superficial. More details on the applied assays and – in particular – their diagnostic accuracy characteristics would be desirable for the reader in order to better estimate the reliability of the diagnostic results.

4.) Materials and methods, sub-heading “Statistical analysis”: The authors state that significance was accepted at p </= 0.05 for all assessments. They should at least explain why correction for multiple testing (e.g., according to Bonferroni-Holmes) was not conducted and how it would have affected the reported significances.

5.) Results: The proportion of patients with AIDS-defining medical conditions is surprisingly high, considering the fact that early diagnosis in line with the 90 : 90 : 90 – rule is aspired in Europe. The authors should at least explain in the discussion why this is the case and comment on the possibility of a selection bias.

6.) Results, table 2: The treatment regiments are quite vaguely described and no stratification by the applied drugs is done. This is regrettable, because details of the therapeutic approach (and also the compliance) might play a role as confounders. The authors should comment on this in the discussion and also in the limitations paragraph.

7.) Discussion, line 221: Next to stratifying by CD4+-counts at a 500 cells / µL breakpoint, it would also be interesting to stratify by differences in the low CD4+ cell ranges.

8.) General minor comment: Although I am not a native speaker myself, I feel that the manuscript would benefit from language proof reading by a native speaker, either arranged by the authors or by the journal.

Author Response

Thank you very much for the substantive review. We took into consideration every suggestion and attempted to reply to all of them.

1) It's true that in our study group the population of late presenters was huge. Among 68 patients there were 46 late presenters and among them 25 people with AIDS-defining diseases. However, in our study group there were also 8 patients with acute HIV infection and 14 individuals who were not late presenters and were not diagnosed at the point of acute HIV infection. We believe that the change in the title in late presenters' favour would diminish the importance of other groups of HIV-infected patients, which are also analyzed in the manuscript. We also think that including all groups of patients in the title would extend the title too much, therefore we decided not to change the current title.

2) The abstract was rephrased and it does not contain more than one p value associated with one marker.

3) In section 2.2 Assessments we added more information about the performed assays of CD4 cell count and HIV viral load, including suppliers, antibodies, equipment used and diagnostic accuracy.

4) We performed an analysis of variance for repeated measures with multiple factors for ΔCD4 cell count, ΔCD4:CD8 ratio, CD4 cell count normalization and CD4:CD8 ratio normalization. A generalized linear model with repeated measures showing the relation between ΔCD4 cell count, ΔCD4:CD8 ratio, CD4 cell count normalization and CD4:CD8 ratio normalization and confounding factors was also performed. The appropiate changes were done in the text and Table 5 with statistics was added.

5) We explained the high proportion of patients with AIDS-defining medical conditions in the discussion at point 4.2. The impact of baseline CD4 count on CD4 recovery.

6) We described the reasons of not performing the stratification by the applied drugs in the discussion. We also included that fact in the limitations paragraph.

7) We added the stratification by lower CD4 cell ranges in the discussion.

8) The whole manuscript underwent some stylistic changes according to the suggestions of a native speaker.

Round 2

Reviewer 1 Report

The authors revised and improved the manuscript following the suggestions. However some major revisions are still needed:

Major revision

1) The introduction is really poor. Authors should better describe the mechanisms that led them to conduct this study.

2) The generalized linear model should also provide readers with indications on the trend shown by the parameters over time. However, no reference to this is present in the results.

3) For the qualitative variables in table 5, the reference category to which the estimated b coefficient is to be linked must be indicated. Example: for the gender variable, is b = 0.177 for females versus males or for males versus females?

 Minor revision

1) The description of the generalized linear model with repeated measures used should be moved to the paragraph “statistical analysis” and only recalled in the results.

2) Tables 3, 6, 8 do not indicate the range but the standard deviation (in round brackets) if the variable had a normal distribution or the interquartile range (in square brackets) if the variable had no normal distribution.

Author Response

Thank you for your review.

Major revision

1. We provided the enhanced description of the mechanisms which led us to conduct this study in the introduction.

2. We added Figure 4 with the results of generalized linear model with repeated measures over time. Below Figure 4 there are also the results for quantitive variables, which cannot be shown in the figure.

3. Above Table 5 we described the reference category for estimated b coefficient.

Minor revision

1. The description of the generalized linear model with repeated measures was moved to the paragraph “statistical analysis”.

2. We adopted the parameter ΔCD4 cell count which signifies the difference in baseline CD4 count and CD4 count after 5 years of sustained antiretroviral therapy and ΔCD4:CD8 ratio which signifies the difference in baseline CD4:CD8 ratio and CD4:CD8 ratio after 5 years of ART. In tables 3, 6, 8 we presented mean value of ΔCD4 cell count and ΔCD4:CD8 ratio and in brackets we showed minimum and maximum value (range) of ΔCD4 cell count and ΔCD4:CD8 ratio. Symbol minus (-) before ΔCD4 or ΔCD4:CD8 means that CD4 cell count or CD4:CD8 ratio after 5 years of ART was lower than baseline CD4 cell count or baseline CD4:CD8 ratio. The description of adopted parameters ΔCD4 cell count and ΔCD4:CD8 ratio are described in the paragraph "Assessments". If necessary, we can change those tables and present standard deviation or interquartile range.

Reviewer 2 Report

Lembas and colleagues have clearly improved the manuscript, including the additional Table 5 for the statistical model. However, the revised version of the manuscript still lacks the information on the instrumentation that was used for flow cytometry as well as the gating strategy. This information is critical to analyze and compare the findings of the current with those of other studies.

Ref 26, the date the ref was accessed seems incorrect as it is in the future, Oct 21, 2022.

Author Response

Thank you for your review. We described in detail the instrumentation and gating strategy for flow cytometry. The paragraph in Materials and Methods (Assessments) was changed.

We are very sorry for the mistake in Ref 26. It was changed to September 2022.

Round 3

Reviewer 1 Report

Minor revision

Tables 3, 6 and 8 should show the mean and standard deviation (in round brackets) if the variable had normal distribution or the median and interquartile range (in square brackets) if the variable had no normal distribution.

Author Response

Thank you for the review. We changed tables 3, 6 and 8 and they now show the mean and standard deviation (in round brackets) if the variable had normal distribution or the median and interquartile range (in square brackets) if the variable had no normal distribution. We also commented on that in the Materials and Methods section (paragraph 'Assessments').